# Detection of Avian Leukosis Virus Subgroup J (ALV-J) Using RAA and CRISPR-Cas13a Combined with Fluorescence and Lateral Flow Assay

**DOI:** 10.3390/ijms251910780

**Published:** 2024-10-07

**Authors:** Shutao Chen, Yuhang Li, Ruyu Liao, Cheng Liu, Xinyi Zhou, Haiwei Wang, Qigui Wang, Xi Lan

**Affiliations:** 1Chongqing Key Laboratory of Herbivore Science, College of Animal Science and Technology, Southwest University, Chongqing 400700, China; cst16634254665@163.com (S.C.); lyh1529824967@163.com (Y.L.); 18717029902@163.com (R.L.); 19923011798@163.com (C.L.); a524448729@163.com (X.Z.); 2Chongqing Academy of Animal Science, Chongqing 402460, China; wahwe@163.com (H.W.); wangqigui@hotmail.com (Q.W.)

**Keywords:** CRISPR-Cas13a, RAA, ALV-J, lateral flow, fluorescence detection

## Abstract

Avian Leukosis Virus (ALV) is a retrovirus that induces immunosuppression and tumor formation in poultry, posing a significant threat to the poultry industry. Currently, there are no effective vaccines or treatments for ALV. Therefore, the early diagnosis of infected flocks and farm sanitation are crucial for controlling outbreaks of this disease. To address the limitations of traditional diagnostic methods, which require sophisticated equipment and skilled personnel, a dual-tube detection method for ALV-J based on reverse transcription isothermal amplification (RAA) and the CRISPR-Cas13a system has been developed. This method offers the advantages of high sensitivity, specificity, and rapidity; it is capable of detecting virus concentrations as low as 5.4 × 10^0^ copies/μL without cross-reactivity with other avian viruses, with a total testing time not exceeding 85 min. The system was applied to 429 clinical samples, resulting in a positivity rate of 15.2% for CRISPR-Cas13a, which was higher than the 14.7% detected by PCR and 14.2% by ELISA, indicating superior detection capability and consistency. Furthermore, the dual-tube RAA-CRISPR detection system provides visually interpretable results, making it suitable for on-site diagnosis in remote farms lacking laboratory facilities. In conclusion, the proposed ALV-J detection method, characterized by its high sensitivity, specificity, and convenience, is expected to be a vital technology for purification efforts against ALV-J.

## 1. Introduction

### 1.1. Overview of Avian Leukosis

Avian leukosis virus (ALV) belongs to the family Retroviridae and the genus Alpharetrovirus. It is characterized by sporadic outbreaks with an incidence rate of 3% to 25%. In 1911, ALV was first discovered by Rous in the United Kingdom, and since then, ALV has been detected in poultry farms worldwide [1,2,3]. Exogenous ALV infection can cause immunosuppressive diseases in chickens, hindering their growth and inducing tumors that may lead to mortality rates of 1% to 2%. In certain poorly managed poultry farms, this mortality rate can even reach as high as 20% [4]. Despite more than a century of research, there is still no vaccine available to protect chickens from ALV infection.

Based on host range, interactions between strains, and antigenic properties of viral envelope glycoproteins, ALV is classified into 11 subgroups (A–K), with subgroup E being endogenous ALV that is non-pathogenic or minimally pathogenic. In 1991, Payne discovered and isolated a new subgroup of avian leukosis virus, naming it subgroup J [5]. Subgroup J can induce various types of tumors in chickens, primarily myeloid tumors [5]. The infectivity and pathogenicity of ALV-J are significantly higher than those of the classical subgroups A, B, C, and D. By the mid-1990s, ALV-J had spread to almost all countries’ broiler chicken populations, causing significant losses to the global poultry industry [6]. ALV-J can be transmitted both horizontally and vertically [7]. If a breeding farm becomes infected with ALV-J and is not promptly sanitized, all parent and commercial flocks may become infected. This situation can lead to the rapid spread of the virus throughout the flock, causing widespread infection. With the eradication of ALV in broilers and layers, ALV-J has basically disappeared from commercial chickens, but recent years have seen serious infections in local breeds and individual farms in China [8]. Therefore, the timely detection and eradication of ALV-J are crucial for ensuring the health of chicken flocks and maintaining production efficiency.

Given that ALV-J infections typically break out and spread in local breeding farms and large-scale poultry farms, the early diagnosis of ALV-J is particularly important for prevention and control. Existing ALV-J detection methods include polymerase chain reaction (PCR) [9], enzyme-linked immunosorbent assay (ELISA), indirect fluorescent antibody reaction (IFA), virus isolation, specific antibody immunohistochemistry, and nucleic acid techniques. Virus isolation is recognized as the international gold standard for virus detection, but it has a long detection cycle and high cost, making it unsuitable for on-site use in farms. These methods have their advantages and disadvantages, but most require complex experimental procedures, expensive equipment, and skilled technicians, making large-scale application in remote areas difficult. Therefore, developing a virus detection method that is highly sensitive and specific, rapid, and easy to operate is crucial for ALV-J detection in remote areas.

### 1.2. CRISPR-Cas Detection System

CRISPR stands for Clustered Regularly Interspaced Short Palindromic Repeats [10]. The CRISPR-Cas system is an adaptive immune system in archaea, first discovered in 1987 [11]. The CRISPR system is differentiated into two primary categories based on its underlying components and functional mechanisms [12]: Type 1 systems comprise multi-protein effector complexes, while Type 2 systems are characterized by a single effector protein. Type 1 systems encompass Subtypes I, III, and IV [13], whereas Type 2 systems include Subtypes II, V, and VI [14]. As of now, more than 26 subtypes have been recognized, with the continuous expansion and differentiation of various types and subtypes. Among Type 2 systems, RNA-guided nucleases Cas12a and Cas13a have been effectively employed as biosensors, targeting single-stranded or double-stranded DNA and single-stranded RNA, respectively [15]. Cas13 forms a complex with crRNA to recognize and target ssRNA, thereby activating the HEPN domain of the Cas13a protein, which can cleave all ssRNA in the environment.

Due to the specific cleavage function and convenience of CRISPR-Cas13a detection, this system has been applied to the detection of various diseases and biomarkers. Zheng Feng et al. developed a SHERLOCK system for target nucleic acid detection using the signal transduction process of CRISPR-Cas13a, specifically detecting the Zika virus [16]. crRNA determines the specificity of CRISPR detection. Parinaz Fozouni et al. applied crRNA targeting to multiple regions of SARS-CoV-2 viral RNA, improving the sensitivity of CRISPR detection [17]. Additionally, lateral flow assays based on the CRISPR-Cas13a system have been used for the serological detection of Nipah virus (NiV) and avian adenovirus [18,19]. To address background interference during RNA amplification, Parinaz Fozouni et al. developed a CRISPR-Cas13a detection method that requires no amplification, using a mobile phone microscope to read results within 30 min, making it more convenient and faster [17]. Compared to traditional PCR and qPCR molecular diagnostic methods, CRISPR methods show advantages in sensitivity, ease of operation, and cost-effectiveness.

Given the successful application of the CRISPR-Cas13a system in the detection of various viruses, we aim to establish a CRISPR-Cas13a system capable of efficiently and specifically detecting ALV-J. By designing specific crRNA, the CRISPR-Cas13a system can recognize and cleave probe RNA, providing a rapid, sensitive, and cost-effective detection method [20]. This not only helps in the timely detection and control of ALV-J spread, but also significantly reduces detection costs, making it especially suitable for resource-limited remote areas.

In this study, we developed a dual-tube detection method, using the CRISPR system to detect ALV-J. We targeted the crRNA of the gp85 gene in the env sequence of the entire ALV-J genome to maximize specificity across all ALV-J isolates or variants.

## 2. Materials and Methods

### 2.1. Design of RAA Primers and crRNA

To improve the specificity of RAA detection, we designed primers targeting the gp85 sequence of the ALV subgroups, as this sequence is fundamental for ALV subgroup identification. First, we downloaded the gp85 sequences of ALV-J, A, B, and K subgroups from the National Center for Biotechnology Information (NCBI, https://www.ncbi.nlm.nih.gov/; accessed on 20 July 2024). Then, we used DNAstar software (version 11.1.0) for sequence alignment to identify the conserved sequences of ALV-J (Figure 1). Based on these conserved sequences, we designed RAA primers with T7 promoters using a primer (https://blast.ncbi.nlm.nih.gov/; accessed on 20 July 2024). All primers were synthesized by Wuhan Tianyi Huayu Gene Technology Co., Ltd. (Wuhan, China).

Based on the aligned conserved sequences of ALV-J, we designed the crRNA for detection on the CHOPCHOP website (CHOPCHOP (uib.no)) (Figure 2). The crRNA was generated from the DNA template via in vitro transcription (IVT) [21], with a T7 polymerase promoter sequence (TAATACGACTCACTATAGGG) added to the 5′ end of the template. The DNA templates for the crRNA were synthesized by Wuhan Tianyi Huayu Gene Technology Co., Ltd.

### 2.2. Preparation of crRNA and RNA Reporter

crRNA is typically generated through in vitro transcription (IVT) from a DNA template. To produce the DNA template for crRNA, DNA oligonucleotides containing the T7 promoter were annealed using a DNA annealing buffer (TAKARA, Changchun, China) to form double-stranded DNA (dsDNA). The synthesized dsDNA was then purified using a DNA purification kit (TAKARA, Changchun, China) and transcribed into RNA using the HiScribe T7 Quick High-Yield RNA Synthesis Kit (New England Biolabs, NEB, Ipswich, MA, USA). The transcribed crRNA was subsequently purified using an RNA purification kit (Sangon Biotech, Shanghai, China). RNA concentration was measured using a NanoDrop One, and the purified crRNA was stored at −80 °C until use.

The fluorescent Reporter RNA labeled with FAM and BHQ1 at both ends was synthesized by Guangzhou Editgene Co., Ltd., Guangzhou, China. The Reporter RNA labeled with FAM and Biotin at both ends was synthesized by Wuhan Tianyi Huayu Gene Technology Co., Ltd., Wuhan, China.

### 2.3. Purification and Large-Scale Expression of LwaCas13a Protein

To express and purify LwaCas13a protein tagged with His and SUMO on a large scale, the plasmid pC013-Twinstrep-SUMO-huLwCas13a (containing the inserted LwaCas13a gene) from Miaolingbio (Wuhan, China) was transformed into *BL21 (DE3) E. coli*. Expression was induced at 16 °C for 14 h by adding 1 mL/L 0.5M IPTG. The induced bacterial culture was sonicated and centrifuged at 8000 rpm for 30 min at 4 °C, and the supernatant containing LwaCas13a was collected and filtered through a 0.22 μm filter. Next, LwaCas13a protein was purified using a pre-packed NI-NTA gravity column. Impurities were eluted using an imidazole gradient of 20 mM, 50 mM, and 80 mM, followed by elution of Cas13a protein bound to the NI column using 500 mM imidazole. The solution containing LwaCas13a protein was collected. To remove the SUMO tag, the protein solution was treated with 1.5% SUMO protease at 4 °C overnight. The protein solution, after SUMO tag removal, was concentrated using ultrafiltration tubes (Millipore, Munich, Germany). The purity of the purified LwaCas13a protein was assessed using SDS-PAGE and Coomassie Brilliant Blue staining, and protein quantification was performed using the BCA Protein Assay Kit (Beyotime, Shanghai, China). Finally, the purified LwaCas13a protein was stored at −80 °C until use.

### 2.4. Virus Preparation

At a poultry farm in Chongqing, we collected liver, plasma, and cloacal samples from birds exhibiting symptoms of avian leukosis. Virus isolation and identification results provided by the Avian Leukosis Laboratory of the College of Animal Science and Technology at Southwest University indicated that these positive samples contained the ALV-J virus. We successfully extracted viral RNA according to the manufacturer’s instructions for the MagicPure^®^ Simple Viral DNA/RNA Kit (TransGen Biotech Co., Ltd., Beijing, China). Subsequently, the viral RNA, dNTPs, reverse transcriptase, and DEPC water were combined to prepare a premix, which was incubated at 85 °C for 15 min to generate cDNA for subsequent experiments. Meanwhile, the RNA samples were securely stored at −80 °C until further use.

The viral strains used to verify the specificity of CRISPR-Cas13a detection for ALV-J were sourced from the Poultry Leukosis Laboratory repository at the College of Animal Science and Technology, Southwest University.

### 2.5. Construction of Standard Plasmids

To evaluate the sensitivity of the detection system for the virus, PCR kits were used to amplify the target gene from the gp85 sequence template (New England Biolabs, NEB, Ipswich, MA, USA). The resulting PCR products were purified using a DNA purification kit (TianGen, Beijing, China). Subsequently, a plasmid vector (PUC-57) was selected and digested with restriction endonucleases (New England Biolabs, NEB, Ipswich, MA, USA), followed by further purification to remove the enzymes. The purified plasmid and insert fragment were mixed in a molar ratio, and subjected to a ligation reaction using T4 DNA ligase. After ligation, the product was transformed into competent Escherichia coli (TOP10) using a transformation kit. Following heat shock and recovery, cells were plated on LB agar containing antibiotics, and incubated overnight to select positive colonies. Finally, plasmid DNA was extracted and validated (using Axygen) to confirm the successful construction of the insert fragment.

### 2.6. Two-Step RAA-CRISPR-Cas13a Detection

In this study, a two-step RAA-CRISPR-Cas13a detection technique was employed, including an RAA amplification step and an LwaCas13a detection system containing T7 RNA polymerase (Figure 3). The specific steps are as follows:

#### 2.6.1. RAA Amplification Step

Following the manufacturer’s instructions (Zhong Ce, Hangzhou, China), a premix was prepared on ice. This premix contained 2.5 μL of forward primer (10 μM), 2.5 μL of reverse primer (10 μM), 2.5 μL of RAA buffer, and 8.65 μL of DEPC water. Next, 40 μL of RNA premix was added to each individual precipitate sample, and the mixture was carefully resuspended on ice. Then, 2.5 μL of Solution B was added to the reaction, and the mixture was quickly vortexed to ensure uniformity. Finally, the reaction was incubated at 42 °C for 30 min in a metal bath.

#### 2.6.2. Step 2: CRISPR-Cas13a Detection

T7 transcription is a critical step in the CRISPR-Cas13a detection process. The reaction mixture was prepared by combining 2 μL of 45 nM LwaCas13a, 1.25 μL of 125 nM fluorescent reporter, 1 μL of RNase inhibitor, 0.18 μL of 1 mM dNTP, 0.5 μL of T7 RNA polymerase (NEB), 0.18 μL of HEPES solution, 0.2 μL of MgCl_2_, 0.6 μL of RAA reaction product, and 14.19 μL of DEPC water. Subsequently, 0.2 μL of 22.5 nM crRNA was added to the mixture to activate the Cas13a protein’s activity. To monitor the reaction, the mixture was incubated at 37 °C, with fluorescence values recorded every 30 s for a total duration of 30 min to collect data.

## 3. Results

### 3.1. Purification of LwaCas13a Protein

Under 21 °C conditions, the expression was induced by adding 0.5 mM IPTG for 16 h, after which we eluted the fusion protein using 500 mM imidazole. Subsequently, the protein was analyzed by SDS-PAGE gel electrophoresis. The solubilized protein samples were concentrated using ultrafiltration and quantified using the BCA Protein Assay Kit. Finally, to remove the SUMO tag, we incubated the protein with SUMO protease at 4 °C overnight (Figure 4).

### 3.2. Feasibility Analysis and Optimization of CRISPR-Cas13a Nucleic Acid Detection

Since the gp85 gene of ALV-J is relatively conserved among isolates, we compared the gp85 sequence of the ALV-J virus preserved in our laboratory with 19 isolates from NCBI to identify potential target sequences. We selected a conserved sequence and designed four pairs of RAA primers. To test the specificity of the RAA primers, we used 2.5% agarose gel electrophoresis to measure the RAA amplified products (Figure 5). The results showed that the brightness of the RAA product of the fourth primer pair was higher under the gel imaging system than that of the other primers.

We compared the activity of five different crRNAs using the CRISPR-Cas13a detection method. The results showed that although the fluorescence value of crRNA2 was significantly higher than that of crRNA4 and the negative control (NC), it was relatively lower compared to that of crRNA1, 3, and 5. The fluorescence value of crRNA3 was significantly higher than that of crRNA1 and crRNA5. Therefore, we chose crRNA3 for subsequent experiments.

To verify the feasibility of the CRISPR-Cas13a trans-cleavage reaction, we used a positive gene as the detection template and tested four different aliquots of the reaction system (Figure 6 and Figure 7). The results showed that when any one of the components (Cas13a protein, crRNA, or RPA amplification product) was missing from the detection system, no fluorescence signal was observed, and the lateral flow assay strip was negative. When Cas13a protein, crRNA, and RPA products were all present, the fluorescence signal significantly increased, and the lateral flow assay strip was positive. The results indicated that only in a complete system can the target RNA of ALV-J activate the Cas13a/crRNA trans-cleavage system.

Since the Cas13a protein, as an enzyme, participates in the detection reaction, the capability of this detection system is closely related to other parameters such as reaction time, temperature, enzyme concentration, and target RNA concentration. To achieve the best detection results, we optimized these reaction conditions. We set the reaction time of the CRISPR-Cas13a system to 10 min, 20 min, 30 min, 50 min, 60 min, and 80 min. The results showed that the fluorescence intensity reached its maximum at 37 °C and 50 min. Therefore, 50 min and 37 °C were selected as the optimal reaction conditions for subsequent experiments. Additionally, we optimized the concentrations of crRNA, Cas13a, and the RAA amplification products. The results indicated that the fluorescence signal reached its maximum when the Cas13a concentration was 100 nM. Finally, 100 nM Cas13a, 1.2 μL RAA product, and 500 nM crRNA were determined to be the optimal reaction conditions for subsequent experiments.

### 3.3. Sensitivity Detection of CRISPR-Cas13a

Next, we explored the detection limit of this system for ALV-J using optimized experimental conditions (Figure 8). We diluted plasmid DNA with an initial concentration of 5.4 × 10^9^ copies/µL in RNase-free water 12 times to create a concentration gradient. This gradient was used to evaluate the sensitivity of RAA-CRISPR-Cas13a for ALV-J detection. The results showed that fluorescence could be observed starting from 20 min. As the concentration of ALV-J increased, more ssRNA was obtained, thus activating more collateral cleavage by Cas13a, leading to a significant increase in fluorescence. A positive signal could be detected when the plasmid DNA concentration was as low as 5.4 × 10^0^ copies/µL, while no fluorescence signal was detected at a concentration of 5.4 × 10^−1^ copies/µL and when the lateral flow assay strip was negative.

### 3.4. Specificity of CRISPR-Cas13a Detection for ALV-J

A qualified avian leukosis virus detection method should be able to specifically recognize the target virus. To evaluate the specificity of this detection system, we selected five different avian viruses for testing. The results, as shown in the figure, indicate that at the same initial concentration (100 nM), the fluorescence intensity changes of different viruses were not significantly different from those of the blank control, except that ALV-J RNA produced a significant signal enhancement (Figure 9). These results indicate that the detection method has good specificity.

### 3.5. Clinical Sample Detection of ALV-J Using CRISPR-Cas13a

To further verify the applicability and accuracy of the CRISPR-Cas13a system for detecting target RNA in complex RNA extracts, we analyzed 429 clinical samples collected from different poultry farms, using CRISPR-Cas13a detection, ELISA detection, and conventional PCR detection (Table 1). The number of positive samples detected by conventional PCR was 63 (positivity rate of 14.7%), the number detected by ELISA was 61 (positivity rate of 14.2%), and the number detected by CRISPR-Cas13a was 68 (positivity rate of 15.9%). The experimental results indicated that the CRISPR-Cas13a-based detection method had the highest positive detection rate for clinical samples. Moreover, the CRISPR-Cas13a detection was faster, more convenient to operate, and more suitable for on-site detection in remote poultry farms.

## 4. Discussion

Infection with ALV-J in poultry leads to reduced production performance, resulting in lower meat yield in broilers and decreased egg production in layers [21]. Additionally, it causes severe immunosuppression, increasing susceptibility to other pathogenic microorganisms, which significantly raises the risk of vaccine failures [22]. Large-scale outbreaks can impose substantial economic losses on poultry farms [23]. Given that there are currently no vaccines or treatments available to prevent or manage ALV-J, the purification of poultry flocks from this virus is particularly critical. The early detection of ALV-J is essential for the timely identification of infected birds, enabling appropriate isolation and treatment measures to mitigate viral spread.

Current diagnostic techniques for ALV-J mainly include qPCR, ELISA, PCR, and virus isolation [24,25,26,27,28]; however, these methods require strict laboratory conditions and specialized equipment, which limits their application for on-site testing. Therefore, it is necessary to develop a convenient detection method that does not require high-tech equipment and can be utilized in non-laboratory settings. In this study, we designed a method for detecting ALV-J based on the CRISPR-Cas13a system combined with RAA isothermal amplification. This approach involves extracting viral RNA from blood samples using a viral nucleic acid extraction kit, followed by 30 min of heat shock. The CRISPR-Cas13a system is then employed to detect target RNA, with results assessed via fluorescence and lateral flow test strips, taking 20 to 30 min for completion. The testing of synthesized plasmids and clinical samples demonstrated that the RAA-CRISPR-Cas13a fluorescence method and lateral flow assay exhibit high sensitivity and specificity, providing intuitive and accurate results within a total duration of 80 min.

Currently, the primary method for detecting ALV-J relies on ELISA, which targets the P27 protein antigen. However, this method may fail to detect the virus during the early stages of infection, and variability in results can arise from different manufacturers and batches of ELISA kits, impacting consistency and reliability. In contrast, the RAA-CRISPR-Cas13a detection method is unaffected by batch variability and allows for the faster completion of tests. Previous studies found that the detection limit of SYBR Green I real-time fluorescence quantitative PCR for ALV-J was 55 copies/μL [29]; a recent study using multiplex qPCR reaction for ALV-J detection showed a detection limit of 13.7 copies/μL [24]. Our study shows that the sensitivity of the RAA-CRISPR-Cas13a detection method is 5.4 × 10^0^ copies/μL, which is higher than that of the RT-qPCR method. Additionally, based on the differences in the gp85 sequence, ALV is divided into ten different subgroups, including endogenous and exogenous ALV. Endogenous ALV is non-pathogenic or has very low pathogenicity, and qPCR detection can result in high non-specific results, i.e., false positives. Previous studies have shown that the low homology between ALV-J and other subtypes is mainly seen in the gp85 region (only 40%), indicating that this region can serve as a specific target for ALV-J detection. Additionally, since the variations in the gp85 gene of ALV-J are mainly concentrated in the hr1, hr2, and vr3 regions, we avoided these three regions when designing RAA primers and crRNA. Therefore, our CRISPR-Cas13a-based detection method improves the sensitivity and specificity of ALV-J detection.

The CRISPR-Cas13a system, as an efficient diagnostic tool, has been used to detect various disease viruses, such as feline herpesvirus and avian influenza virus. In this study [30,31], we validated that the sensitivity of the CRISPR-Cas13a system is 5.4 × 10^0^ copies/μL, with a higher positive conformity rate for blood clinical samples compared to that of PCR and ELISA detection methods. We also found that the virus culture testing of discrepant blood samples yielded positive results. We developed both fluorescence and lateral flow detection methods, which can sensitively and specifically detect ALV-J within 85 min. Notably, the combination of CRISPR-Cas13a with lateral flow assay is very convenient and can be used for rapid on-site detection.

The CRISPR-Cas13a detection method is cost-effective, with an average reaction cost of less than CNY 3, while qPCR reagents can reach up to CNY 6, excluding sampling and transportation fees. Unlike traditional PCR or qPCR methods, the CRISPR-Cas13a method requires no complex equipment or specialized personnel; the entire process is conducted at a constant temperature of 37 °C and yields results in about one hour. Furthermore, the results are visually interpretable, making it ideal for remote poultry farms lacking laboratory facilities. Our study significantly enhances the detection capabilities of poultry farmers, facilitating the early identification and isolation of infected individuals, thus reducing the risk of disease transmission. Additionally, timely detection and intervention can improve overall poultry health, allowing farmers to conduct rapid and accurate testing on site without relying on specialized laboratories.

In summary, we have developed a novel CRISPR-Cas13a detection method for ALV-J that is capable of detecting virus levels as low as 5.4 × 10^0^ copies/μL, with no cross-reactivity to various avian viruses. The total detection time does not exceed 85 min, and the procedure is user-friendly. This research provides a robust tool for the purification and assessment of ALV-J in breeding farms.

## Figures and Tables

**Figure 1 ijms-25-10780-f001:**
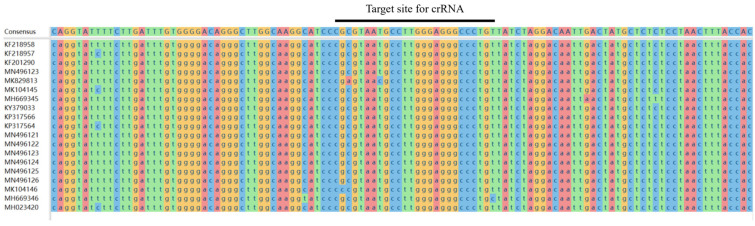
Comparison of different ALV-J gp85 sequences and crRNA target sites.

**Figure 2 ijms-25-10780-f002:**
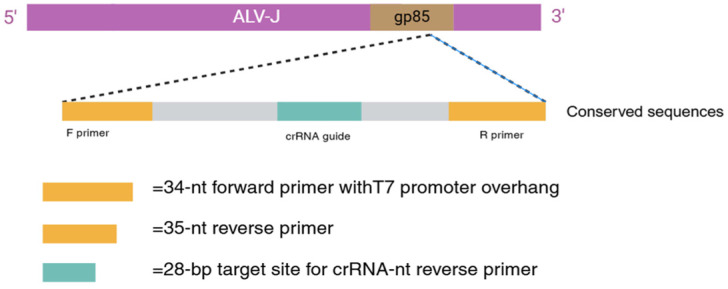
Target sites of crRNA for RAA-CRISPR13a detection and positions of the RAA primer sets.

**Figure 3 ijms-25-10780-f003:**
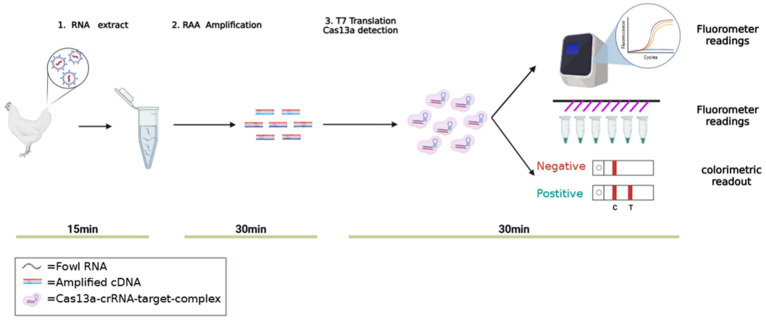
Schematic diagram of the principle for ALV-J detection based on RAA-CRISPR-Cas13a. Genomic material extracted from blood is introduced into the RAA reaction to amplify the target sequence (ssDNA). Subsequently, the obtained ssDNA is transcribed into ssRNA using T7 RNA polymerase. When the crRNA recognizes the target sequence within the ssRNA, it activates the Cas13a enzyme. Concurrently, the activated Cas13a exhibits collateral cleavage activity, which cleaves RNA fluorescence probes. This can be detected through fluorescence detection devices, such as a quantitative PCR machine or an enzyme-linked immunosorbent assay (ELISA) reader, or alternatively through a lateral flow dipstick (LFD).

**Figure 4 ijms-25-10780-f004:**
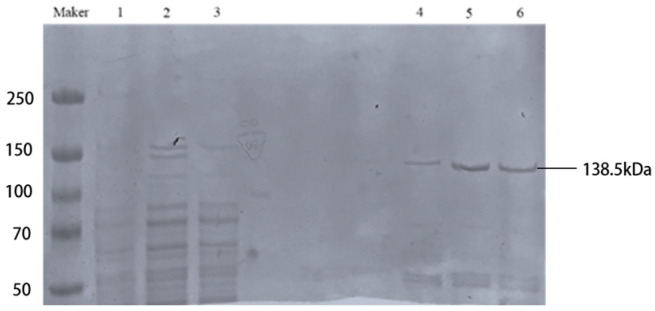
SDS-PAGE electrophoresis for LwaCas13a purification. Lane 1: supernatant after IPTG induction. Lane 2: supernatant after sonication. Lane 3: supernatant after 20–80 mM imidazole elution. Lanes 4, 5, and 6: supernatant after 500 mM imidazole elution, representing Cas13a protein.

**Figure 5 ijms-25-10780-f005:**
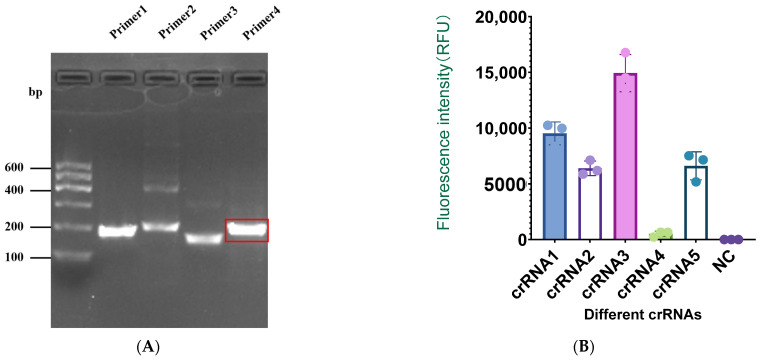
(**A**) Gel electrophoresis of RAA products amplified by primer 1, 2, 3, and 4. (**B**) Identification of highly active crRNA using real-time fluorescence readings.

**Figure 6 ijms-25-10780-f006:**
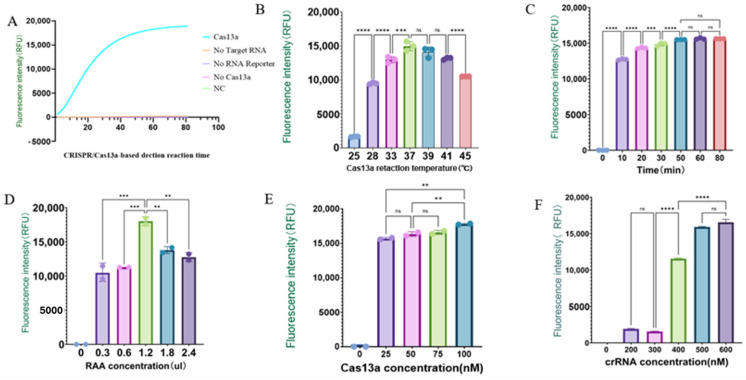
Validation and optimization of the CRISPR-Cas13a detection system: (**A**) Fluorescence signal detected with different components added; (**B**) optimization of reaction temperature from 25, 28, 33, 37, 39, 41, and 45 °C; (**C**) optimization of reaction time from 10, 20, 30, 40, 50, and 60 min; (**D**) optimization of Cas13a protein concentration from 25, 50, 75, and 100 nM; (**E**) optimization of RAA input amount from 0.3, 0.6, 1.2, 1.8, and 2.4 μL; (**F**) optimization of crRNA concentration from 200, 300, 400, 500, and 600 nM. (ns: *p* > 0.05; **: *p* ≤ 0.01; ***: *p* ≤ 0.001; ****: *p* ≤ 0.0001).

**Figure 7 ijms-25-10780-f007:**
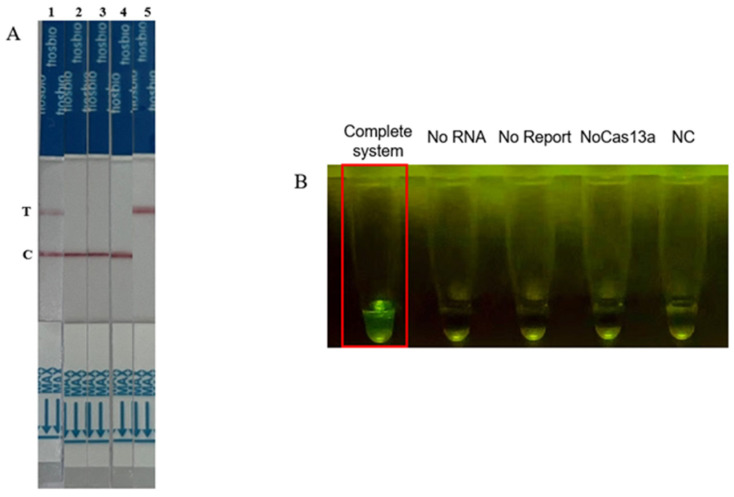
The validation of the RAA-CRISPR-Cas13a system using fluorescence and lateral flow assays. (**A**) The validation of the CRISPR-Cas13a detection system using a lateral flow assay. The test strip has a C line and a T line. 1: all components present; 2: target RNA missing; 3: RNA report missing; 4: Cas13a missing; 5: only water. The result is positive only when both the C line and T line are colored. (**B**) Fluorescence coloration after 30 min of reaction with different components.

**Figure 8 ijms-25-10780-f008:**
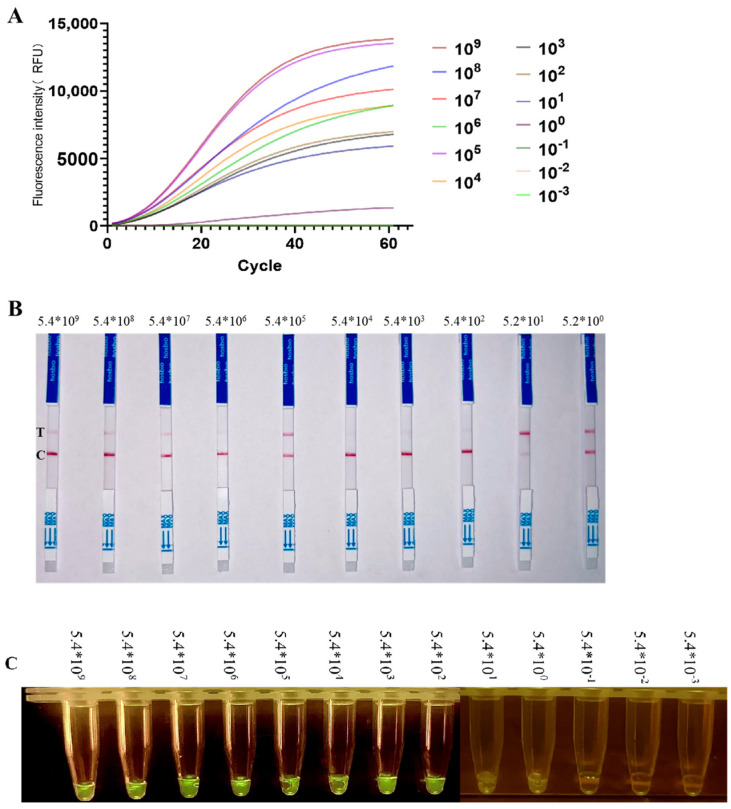
Sensitivity of CRISPR-Cas13a detection. (**A**) Fluorescence values for CRISPR-Cas13a detection of gp85 gradient-diluted plasmid; (**B**) lateral flow assay strips for CRISPR-Cas13a detection of gp85 gradient-diluted plasmid; (**C**) fluorescence coloration after CRISPR-Cas13a detection of gp85 gradient-diluted plasmid.

**Figure 9 ijms-25-10780-f009:**
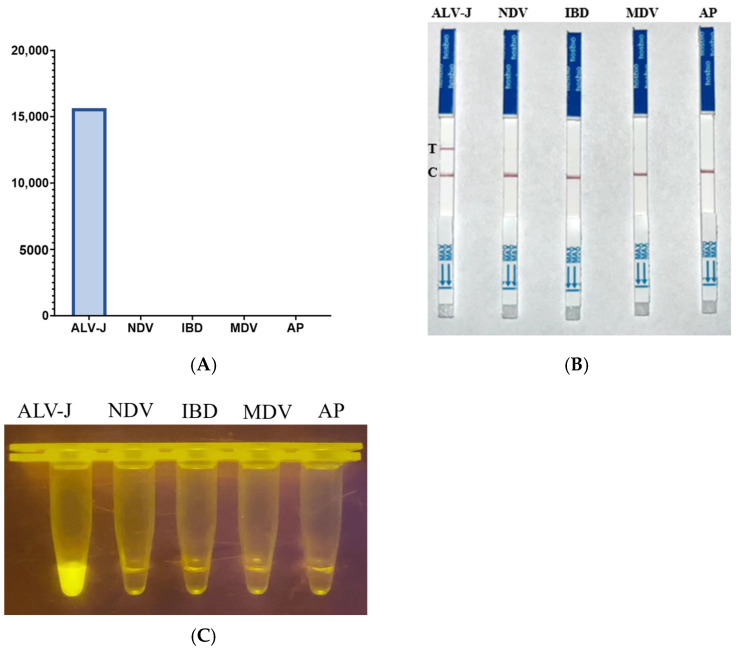
Specificity of CRISPR-Cas13a detection: (**A**) fluorescence values for detecting different avian viruses; (**B**) lateral flow assay for detecting different avian viruses; (**C**) fluorescence coloration for detecting different avian viruses.

**Table 1 ijms-25-10780-t001:** Detection of 429 blood samples using ELISA, PCR, and CRISPR-Cas13a.

Samples	Total	ELISA	PCR	CRISPR/Cas13a
Cloacal	310	30	30	35
Blood	119	31	33	33
Total	429	61	63	68
Positivity Rate	-	14.2%	14.7%	15.9%

## Data Availability

Data is contained within the article or Appendix A.

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
