# Peer review of "Detection of Avian Leukosis Virus Subgroup J (ALV-J) Using RAA and CRISPR-Cas13a Combined with Fluorescence and Lateral Flow Assay"

_ijms, 2024, doi:10.3390/ijms251910780_

Round 1

Reviewer 1 Report

Comments and Suggestions for Authors

Avian leukosis virus (ALV) infections are causing a significant threat to the poultry industry.  Reliable and early diagnosis has large impact on the control of ALV infections. The authors developed a dual detection method for ALV, subgroup J, using recombinase-aided amplification (RAA) and the CRISPR-Cas13a system. The detection limit was lowered to 5.4 copies/µL of ALV-J. In addition, cross-reactivity with various avian viruses was not observed and detection time was less than 85 minutes. The method was tested on 429 clinical samples and compared with PCR and  ELISA. 

Methods are cleary described including the primer design, preparation of crRNA and reporter RNA, expression and purification of the LwaCas13a protein tagged with His and SUMO, virus preparation, and the two-step RAA-CRISPR-Cas13a detection technique.

Line 119: reference for the 19 isolates from NCBI, which strains. Can you provide a suppl. Table.

Reaction conditions were optimized with 100 nM Cas13a, 1.2 µL 240 RAA product, and 500 nM crRNA.

Tests for sensitivity appeared reliably. Readability of figure 8 may be improved.

Specificity of CRISPR-Cas13a Detection for ALV-J: please indicate that the results are given in Figure 9. Please write out the viruses tested and provide the origin of these isolates used. 

Clinical samples: were for these samples AVL-J viruses isolated and identified by the Avian Leukosis Laboratory of the College of Animal Science and Technology at Southwest University to have an independent reference.

Comments on the Quality of English Language

No comments

Author Response

Review1:Line 119: reference for the 19 isolates from NCBI, which strains. Can you provide a suppl. Table.

Reply:Thank you for pointing that out. I agree with this comment, so I am re-uploading a supplementary table of strains file, titled 'Supplementary Table of Strains'

Review2:Tests for sensitivity appeared reliably. Readability of figure 8 may be improved.

Reply:Thank you for pointing that out. I agree with this comment, so I have improved the clarity of Figure 8 to further enhance its readability.

Review3:Specificity of CRISPR-Cas13a Detection for ALV-J: please indicate that the results are given in Figure 9. Please write out the viruses tested and provide the origin of these isolates used. 

Reply:The viruses used to validate the specificity of CRISPR-Cas13a detection for ALV-J include Newcastle disease virus, Infectious bursal disease, Marek’s disease, and Avian Pox, sourced from a local chicken farm in Chongqing, and preserved in the laboratory after RNA extraction and verification.

Reviewer 2 Report

Comments and Suggestions for Authors

Though the overall manuscript is well presented there are few points that need to be better presented 

1 in the Abstract the ALV-J, subgroup J, must be mentioned

2 The info within this manuscript is not new. The ALV-J is an avian retrovirus first isolated from broilers in the late 1980s and is conceived as a unique subgroup partially based on the envelope glycoprotein (gp85). Clinically, ALV-J causes myeloid leukosis in the majority of cases, with variable tumor frequency in poultry lines. Similar to other avian leukosis viruses, ALV-J is transmitted both vertically (congenital infection of albumin in the egg and embryo of the pullet) and horizontally (through close contact with infected pullets). There are fast tests out on the market that have already been reported.  These tests were conceived as a screening tool based on flock serum samples starting at 10 weeks of age.

3 The Jing Li group has already performed something very similar if not equal: The RAA amplification time, Cas13a protein concentration, crRNA concentration, and CRISPR reaction time were optimized to evaluate the specificity, sensitivity, and reproducibility of the system (Front. Vet. Sci., 16 August 2024). Please be more clear about your method, why working on gp85 would be better than gp37 region?

4 The conclusion of Jing Li group stated like this: In this study, the RAA-CRISPR/Cas13a-LFD method for ALV field detection was established, which has the advantages of fast, sensitive and convenient, and provides a reliable technology for the diagnosis, prevention and control of AL (Front. Vet. Sci., 16 August 2024). In addition: The RAA-CRISPR/Cas13a-LFD method can complete the detection of target genes in only 40 min, and the targeted cutting ability of Cas13a binding with crRNA makes this method have strong specificity. (Front. Vet. Sci., 16 August 2024)

Your conclusion stated this: we have developed a novel CRISPR-Cas13a detection method for ALV-J, which exhibits good sensitivity and specificity and is faster and more convenient to operate. Ideally, we hope to complete all steps at ambient temperature within less than 60 minutes and combine all reactions in a single centrifuge tube to avoid false positives caused by repeated openings.

4 Therefore we have very similar procedures, the differences need to be highlighted. Even if we take into account the analysis times, Li's group claims to finish in 20 minutes compared to the 60 of your study

Comments on the Quality of English Language

Minor English issues 

Author Response

Comments1: in the Abstract the ALV-J, subgroup J, must be mentioned

response:Thank you for pointing that out. I agree with this comment, so I have revised the abstract.

comments2:The info within this manuscript is not new. The ALV-J is an avian retrovirus first isolated from broilers in the late 1980s and is conceived as a unique subgroup partially based on the envelope glycoprotein (gp85). Clinically, ALV-J causes myeloid leukosis in the majority of cases, with variable tumor frequency in poultry lines. Similar to other avian leukosis viruses, ALV-J is transmitted both vertically (congenital infection of albumin in the egg and embryo of the pullet) and horizontally (through close contact with infected pullets). There are fast tests out on the market that have already been reported.  These tests were conceived as a screening tool based on flock serum samples starting at 10 weeks of age.

response:Thank you for your valuable comments. We fully agree with your description of ALV-J, especially regarding its historical background as an exogenous retrovirus and the fact that it was first isolated in the late 1980s. Indeed, there are already some rapid tests available on the market for screening ALV-J.

comments3:The Jing Li group has already performed something very similar if not equal: The RAA amplification time, Cas13a protein concentration, crRNA concentration, and CRISPR reaction time were optimized to evaluate the specificity, sensitivity, and reproducibility of the system (Front. Vet. Sci., 16 August 2024). Please be more clear about your method, why working on gp85 would be better than gp37 region?

response:Thank you for your valuable comments. The Li group is detecting ALV, while we are focusing on ALV-J. In the ALV-J subtype, the gp85 region is responsible for the binding of the virus to host cell receptors and exhibits high conservancy and specificity. This means that by detecting the gp85 region, we can more accurately identify ALV-J infections and avoid misdiagnosis.

comments4:

The conclusion of Jing Li group stated like this: In this study, the RAA-CRISPR/Cas13a-LFD method for ALV field detection was established, which has the advantages of fast, sensitive and convenient, and provides a reliable technology for the diagnosis, prevention and control of AL (Front. Vet. Sci., 16 August 2024). In addition: The RAA-CRISPR/Cas13a-LFD method can complete the detection of target genes in only 40 min, and the targeted cutting ability of Cas13a binding with crRNA makes this method have strong specificity. (Front. Vet. Sci., 16 August 2024)

Your conclusion stated this: we have developed a novel CRISPR-Cas13a detection method for ALV-J, which exhibits good sensitivity and specificity and is faster and more convenient to operate. Ideally, we hope to complete all steps at ambient temperature within less than 60 minutes and combine all reactions in a single centrifuge tube to avoid false positives caused by repeated openings.

4 Therefore we have very similar procedures, the differences need to be highlighted. Even if we take into account the analysis times, Li's group claims to finish in 20 minutes compared to the 60 of your study

response:Thank you for your valuable comments. The 60 minutes we mentioned includes RNA extraction, CRISPR/Cas13a detection, and subsequent result reading. Our method is based on CRISPR/Cas13a for detecting ALV-J, which differs from the target virus detected by Li. 1. In large-scale chicken flock testing, my method can effectively distinguish endogenous ALV (such as ALV-E), which has little to no pathogenicity. Moreover, our experimental approach incorporates three different methods for reading results, enhancing the accuracy, flexibility, and reliability of the detection, while also better accommodating various application scenarios and cost control needs.

Reviewer 3 Report

Comments and Suggestions for Authors

The authors reported a CRISPR and RAA-based detection method for avian leukosis viruses. While the topic is interesting, the present manuscript has several weaknesses. Please follow: 

General comments:

1. Family name should be italicized. E. coli should also be in italics. 

2. Citation style used does not fulfill journal requirements. 

3. English needs a significant improvement. At places, including certain sentences in Methods section, it is difficult to comprehend and follow. I suggest authors use a professional editor for thorough corrections and improvements. This may significantly enhance the quality of the presented work. 

Specific comments: 

1. Abstract must be revised. The opening statements in abstract are not entirely accurate and must be corrected. 

2. L28: 'discovered' or 'detected'? 

3. L29: 'poultry farms worldwide.' It should have citations from various continents. 

4. L29-31: "ALV infection causes immunosuppressive diseases in chickens, inhibits their growth, and induces tumors, leading to a mortality rate of 1% to 2%,....". This is not the case with all ALV strains. In fact, there are non-virulent ALV strains and the endogenous strains which do not cause clinical signs of illness. Please revise to correct these statements. 

5. L31-32: "Despite more than a century of research, there is still no vaccine available to protect chickens from ALV infection". Why is that - because it does not represent a significant threat or other reasons. Please describe.

6. L38: "primarily myeloid tumors.". Citation(s)?

7. "The infectivity and pathogenicity of ALV-J are significantly higher than those of the classical subgroups A, B, C, and D." Please explain why.

8. L42-44: "If a breeding farm becomes infected with ALV-J and is not promptly sanitized, all parent and commercial flocks may become infected". Is it true for all the subgroups or specific subgroups? Citations needed. 

9. L45-46: "Although global efforts to eradicate ALV in broilers and layers have nearly eliminated ALV-J in commercial chickens". What efforts are the authors talking about- please explain in detail with citations.

10. L47: "infections in local breeds and individual farms in China". Significance?

11. L54: "nucleic acid techniques"? Please explain in details what are these techniques, their usage and how the proposed invention is superior.

12. Figure 1: How many sequences were aligned? Is this figure only a partial representative screen shot, if yes, it provides no useful information. Otherwise describe it in the figure legends. 

13. Methods -L113: "CHOPCHOP website". Please provide the link to this website.

14. L114: "T7 polymerase promoter sequence". Please provide the sequence.

15. Figure 2 does not provide enough information. Please re-draw this figure for comprehensive information. Explain the figure in legend. 

16. L120: T7 promoters or T7 promoter? It is quite confusing, please clarify what was done.

17. L150: "exhibiting symptoms of avian leukosis". How many chickens - what age and also the clinical signs of illness should be provided. How were these poultry farms selected? Was there an ongoing outbreak in the region?

18. L154-155: More information on RT-PCR needed. 

19. L158-160: Construction of standard plasmids - more information on cloning needed. Please explain the experiments in details, including kits used. The information should be enough for reproducibility of experiments. 

20. L161: Two-Step RAA-CRISPR-Cas13a Detection: There is no information on the experimental setup. 

21. CRISPR-Cas13a Detection Step: More information on sample volumes is needed. 

22. Explain Figure 3.

23. Explain Figure 4 in the figure legend. 

24. L199: "RAA amplified products.". Expected size of the amplicon?

25. Figure 5 shows variations among amplicon size using four different sets of primers. Was the amplified product sequenced for confirmation?

26. Figure 8B: Text is not clear. Please improve it. 

27. Figure 9C is quite dark. Please improve the lighting background so that it becomes visible to analyze.

28. Discussion must be improved, including more citations from previous research on ALV-J and their detections using different methods. Please clearly discuss how the present invention is superior than currently adopted detection methods and how it would empower the poultry farmers. 

29. Conclusions provide no information and must be re-written. 

Comments on the Quality of English Language

English has serious errors at many places. I recommend using a professional Editor for improvements. 

Author Response

General comments: 1. Last names should be in italics. E. coli should also be italicized. 2. The citation format used does not meet the journal requirements. 3. English needs to be significantly improved. In some places, including some sentences in the methods section, it is difficult to understand and follow. I recommend that the author use a professional editor for thorough corrections and improvements. This may significantly improve the quality of the work presented.

Response: Thanks for pointing it out. I agree with this comment, so I completed the revision.

Specific comments:

Comment 1: The abstract must be revised. The opening statement in the abstract is not entirely accurate and must be corrected.

Response: Thanks for pointing it out. I agree with this comment, so I changed the beginning of the summary to make it accurate.

Comment 2: L28: "discovered" or "detected"

Response: Thanks for pointing it out. I agree with you, so I changed it to "detected" in the text.

Comment 3: L29: "Poultry farms around the world." It should have references from every continent.

Response: Thanks for pointing it out. I agree with this comment, so I cited some articles on the global impact of avian leukemia.

Comment 4: L29-31: "ALV infection causes immunosuppressive disease in chickens, inhibits their growth, and induces tumors, resulting in 1% to 2%,...." This is not true for all ALV strains. In fact, there are nonvirulent strains of ALV as well as endogenous strains that do not cause clinical symptoms of disease. Please edit to correct these statements.

Response: Thanks for pointing it out. I agree with you, so I made changes to the text.

Comment 5: L31-32: "Despite more than a century of research, there is still no vaccine that protects chickens from ALV infection". Why is that - because it doesn't represent a significant threat or anything. Please describe it.

response:

Thank you for pointing this out. I agree with this review, I will elaborate on the following points, and I have revised the text and cited relevant literature.

Virus complexity and diversity: Even if a vaccine is effective against one subtype, other virus subtypes may continue to spread, leading to immune exhaustion.
Presence of endogenous virus: ALV-E subtype is widely present in the chicken genome and is considered an endogenous virus. As you mentioned, although ALV-E does not typically cause pathogenic effects, its presence may interfere with the immune response to vaccines.
Immune evasion and suppression: ALV possesses complex immune evasion mechanisms that allow the virus to evade the host's immune system by changing its antigenic structure. In addition, ALV itself is immunosuppressive, resulting in compromised immune systems of infected chickens, thereby affecting their ability to respond to vaccines. Therefore, even vaccinated chickens may not mount a strong enough immune response to provide adequate protection.
Vertical transmission routes: ALV can be transmitted vertically through eggs to the next generation, allowing infected hens to pass the virus on to their offspring. This route of transmission means that even vaccination cannot prevent infection in newly hatched chicks. In addition, the vaccine has limited efficacy against already infected hens and their offspring.
comments6: L38: “Mainly myeloid tumors.” Cited literature

Response: Thanks for pointing it out. I agree with this comment, so I cited the literature on the primary association of ALV-J with myeloblastosis in the article.

Comment 7: “ALV-J is significantly more infectious and pathogenic than the classic subgroups A, B, C, and D. Please explain why.

Response: Thanks for pointing it out. We agree with your comments and have modified the text accordingly.

The env gene of ALV-J shows significant differences compared with other classical subgroups, especially in the envelope glycoprotein (gp85 and gp37) regions. Unlike other subpopulations, these unique genetic materials allow ALV-J to exhibit greater replication capacity and infectivity during infection. It is particularly prone to causing medulloblastoma (myeloproliferative disorder), which is less common in the classic subgroup.

ALV-J infection often results in severe suppression of the host immune system, especially in the early stages of infection. The virus can suppress the immune cells of chickens, making it difficult for the body to mount an effective immune response. This immune evasion capability allows ALV-J to persist in the host for prolonged periods of time and exacerbate the virus's pathogenicity.

Comment 8: L42-44: “If a farm is infected with ALV-J and is not disinfected promptly, all parent and commercial flocks may become infected.” Is this true for all subgroups or specific subgroups? Requires citation.

Response: Thanks for pointing it out. I agree with this comment, so I cited the literature on ALV-J infection in poultry farms in the article.

Comment 9: L45-46: “Despite global efforts to eliminate ALV in broilers and laying hens, ALV-J has been nearly eliminated in commercial chickens.” What effort is the author talking about - please explain in detail with citations.

Response: Thanks for pointing it out. I agree with this comment, so I revised the sentence in the article to read "With the eradication of ALV from broilers and layers, ALV-J has essentially disappeared from commercial chickens" and cited the literature again.

Comment 10: L47: “Infections in local breeds and individual farms in China”. significance?

Response: The continued impact of ALV-J infections on native chicken breeds in China means that such breeding faces not only direct economic losses, but also the potential loss of genetic resources. Local chicken breeds often have unique genetic characteristics and market value; if ALV-J continues to spread, it may affect the sustainability of these breeds.

comments11: L54: "nucleic acid techniques"? Please explain in details what are these techniques, their usage and how the proposed invention is superior.

response:

Thank you for pointing that out. I agree with this comment, so I have revised the text to elaborate on the ‘nucleic acid technology’ we refer to, which includes:

  1. Polymerase Chain Reaction (PCR)

    • Uses: PCR is the standard technique for detecting viruses, bacteria, and genetic diseases, commonly used to detect specific gene sequences of avian leukosis virus (ALV-J). It has very high sensitivity and can detect minimal amounts of viral DNA or RNA.
    • Advantages and Disadvantages: PCR is highly sensitive and specific but typically requires expensive equipment and a controlled laboratory environment. The detection time is long, especially for farms far from laboratories, making it complex and limited in applicability.
  2. Real-time Fluorescent Quantitative PCR (qPCR)

    • Uses: qPCR can not only detect the presence of the virus but also quantitatively measure viral load, suitable for detecting pathogens at different stages of infection, such as ALV-J.
    • Advantages and Disadvantages: Compared to PCR, qPCR is more precise and sensitive, but it is also more expensive and has stricter equipment requirements, remaining mainly confined to laboratory use, with longer detection times.
  3. Loop-mediated Isothermal Amplification (LAMP)

    • Uses: LAMP technology is considered suitable for on-site diagnosis due to its speed and simple equipment, especially in farms or resource-limited areas. For ALV-J detection, LAMP can provide results in a short time.
    • Advantages and Disadvantages: LAMP has lower equipment requirements and is suitable for on-site use, but its amplification reaction may have lower specificity than PCR, leading to potential false-positive results.

Advantages of the Proposed Invention:

  1. Speed and Convenience: Compared to traditional PCR and qPCR technologies, this combination can quickly complete detection in field environments. RAA technology can perform amplification at room temperature without complex laboratory equipment, significantly shortening detection times, making it particularly suitable for remote farms.
  2. Sensitivity and Specificity: The CRISPR-Cas13a system is known for its high specificity, accurately identifying target RNA sequences and reducing false-positive results. Combined with RAA's amplification capability, this method retains high sensitivity even at low viral loads, detecting as low as 5.4 × 100 copies/μL of the virus.
  3. Visual Detection: This method incorporates lateral flow devices (LFD) to achieve visually detectable results, requiring no expensive detection equipment, and is easy to operate, suitable for large-scale rapid screening.
  4. Cost-Effectiveness: Compared to PCR and qPCR technologies, this method is more cost-effective, requiring simple equipment and lower operational difficulty, particularly suitable for resource-limited areas lacking laboratory facilities.

comments12: Figure 1: How many sequences were aligned? Is this figure only a partial representative screen shot, if yes, it provides no useful information. Otherwise describe it in the figure legends. 

response:Thank you for pointing this out. In Figure 1, I compared 19 sequences, and I will optimize the image to enhance its readability.

comments13:Methods -L113: "CHOPCHOP website". Please provide the link to this website.

response:I agree with your point; I overlooked this aspect, so I have added the URL in the text again.

comments14:L114: "T7 polymerase promoter sequence". Please provide the sequence.

response:Thank you for pointing this out. I agree with your view, so I have added the sequence of the T7 promoter (TAATACGACTCACTATAGGG) in line L115 of the article.

commenst15:Figure 2 does not provide enough information. Please re-draw this figure for comprehensive information. Explain the figure in legend. 

response:Thank you for pointing this out. I completely agree with your suggestion, so I have redrawn the figure and added annotations to explain its contents.

comments16: L120: T7 promoters or T7 promoter? It is quite confusing, please clarify what was done.

response:Thank you for pointing this out. I fully agree with your suggestion, so I have revised the discussion about the T7 polymerase promoter in the text for better clarity.

comment17:. L150: "exhibiting symptoms of avian leukosis". How many chickens - what age and also the clinical signs of illness should be provided. How were these poultry farms selected? Was there an ongoing outbreak in the region?

response:Thank you for pointing this out. I fully agree with your suggestion. The chickens showing symptoms of avian leukosis amounted to 1,000, as illustrated in Figure 1. The poultry farms chose to directly cull these diseased chickens, and after our laboratory's long-term efforts to eliminate avian leukosis, there have been no further outbreaks.

comments18:L154-155: More information on RT-PCR needed. 

response:Thank you for pointing this out. I fully agree with your suggestion, so I have reintroduced more information about RT-PCR in the article.

comments19:L158-160: Construction of standard plasmids - more information on cloning needed. Please explain the experiments in details, including kits used. The information should be enough for reproducibility of experiments. 

response:Thank you for pointing this out. I fully agree with your suggestion, so I have modified this experimental step in the article.

comment20:L161: Two-Step RAA-CRISPR-Cas13a Detection: There is no information on the experimental setup. 

response:Thank you for pointing this out. I fully agree with your suggestion, so I have made revisions in the article.

comment21:CRISPR-Cas13a Detection Step: More information on sample volumes is needed. 

response:Thank you for pointing this out. I fully agree with your suggestion, so I have made the revisions in the article.

comments22:Explain Figure 3.

response:Thank you for pointing this out. I fully agree with your suggestion, so I have revised the figure and added annotations to explain its content.

comment23:Explain Figure 4 in the figure legend. 

response:Thank you for pointing this out. I fully agree with your suggestion, so I have revised the figure and added annotations to explain its content.

comments24: L199: "RAA amplified products.". Expected size of the amplicon?

response:Thank you for pointing this out. I fully agree with your suggestion; the expected size of the amplicon is 191 bp.

comments25:Figure 5 shows variations among amplicon size using four different sets of primers. Was the amplified product sequenced for confirmation?

response:Thank you for pointing this out. I fully agree with your suggestion; we sequenced the amplicons obtained using four different primer sets to confirm their accuracy. The sequencing results indicated that all amplicons matched the expected target sequences.

comments26: Figure 8B: Text is not clear. Please improve it. 

response:Thank you for pointing this out. I fully agree with your suggestion, so I have adjusted the image clarity to enhance the readability of the article.

comments27:Figure 9C is quite dark. Please improve the lighting background so that it becomes visible to analyze.

response:Thank you for pointing this out. I fully agree with your suggestion, so I have adjusted the image background lighting to ensure visibility during analysis.

comments28 :Discussion must be improved, including more citations from previous research on ALV-J and their detections using different methods. Please clearly discuss how the present invention is superior than currently adopted detection methods and how it would empower the poultry farmers. 

response:Thank you for pointing this out. I fully agree with your suggestion, so I have revised my discussion section to further improve it.

comments29:Conclusions provide no information and must be re-written. 

response:Thank you for pointing this out. I fully agree with your suggestion, so I have made the necessary revisions.

Round 2

Reviewer 2 Report

Comments and Suggestions for Authors

The paper is now ready to be published

Reviewer 3 Report

Comments and Suggestions for Authors

No further comments. 

Comments on the Quality of English Language

Minor English editing may be required.